



# Could promontories have restricted sea-glacier penetration into marine embayments during Snowball Earth events?

Adam J. Campbell[1,3], Betzalel Massarano[1,4], Edwin D. Waddington[1], Stephen G. Warren[1,2]

1 Department of Earth and Space Sciences, University of Washington, Seattle, Washington, USA

5    2 Astrobiology Program, University of Washington, Seattle, Washington, USA

3 now at School of Surveying, University of Otago, Dunedin, New Zealand

4 now at Pacific Science Center, Seattle, Washington, USA

*Correspondence to*: Adam Campbell (a.campbell@otago.ac.nz)

**Abstract.** During the Neoproterozoic, Earth experienced several climate excursions of extreme cold, often referred to as the
Snowball Earth events. During these periods, thick flowing ice, referred to as sea glaciers, covered the entire planet's oceans. In addition, there is evidence that photosynthetic eukaryotic algae survived during these periods. With thick sea glaciers covering the oceans, it is uncertain where these organisms survived. One hypothesis is that these algae survived in marine embayments hydrologically connected to the global ocean, where the flow of sea glacier could be resisted. In order for an embayment to act as a refugium, the invading sea glacier must not completely penetrate the embayment. Recent
studies have shown that straight-sided, marine embayments could have prevented full sea-glacier penetration under a narrow range of climate conditions suitable for the Snowball Earth events. Here we test whether promontories, i.e. headlands emerging from a side shoreline, could further restrict sea-glacier flow. We use an ice-flow model, suitable for floating ice, to determine the flow of an invading sea glacier. We show that promontories can expand the range of climate conditions allowing refugia by resisting the flow of invading sea glaciers.

**1 Introduction**

During the Neoproterozoic, the entire upper ocean surface may have been covered by ice several hundreds of meters thick [*Goodman and Pierrehumbert*, 2003; *Goodman*, 2006; *Li and Pierrehumbert*, 2011]. These periods of global ocean ice cover, commonly referred to as the Snowball Earth events, lasted multiple millions years during at least two separate periods of the Neoproterozoic [*Hoffman et al.*, 2008]. Fossil evidence demonstrates that photosynthetic eukaryotic algae existed
immediately prior to and after these events [*Cohen et al.*, 2015, *Knoll*, 1992; *Macdonald et al.*, 2010], implying they survived during the Snowball Earth events. Ice covering the ocean, thick enough to deform under its own weight as a "sea glacier," restricted the possible locations for survival of photosynthetic organisms. However, sea-glacier ice would have been





too thick to allow for the transmission of light to the liquid water below the sea glacier, a necessary condition for photosynthesis, thereby setting up an apparent contradiction. It has been suggested that the evidence of survival photosynthetic organisms during these times implies an incomplete cover of ice. *Hoffman and Schrag* [2000] suggested that small pools of open water could be maintained near volcanic islands, in order to reconcile the apparent contradiction; however this theory has not been rigorously explored. *Warren et al.* [2002] mentioned that marine embayments may have provide refugia for photosynthetic algae during the Snowball Earth events; as a sea glacier encountered a marine embayment the sea-glacier invasion would be resisted by drag along the walls of the embayment. If such a marine embayment had appropriate geometry, the invading sea glacier might be unable to fully penetrate the embayment. The portion of the embayment not penetrated by the sea glacier might then be able to provide a refugium for photosynthetic organisms. In an exploration of this idea, Campbell et al. [2011, 2014] demonstrated that, under a range of climate conditions that could be expected during the Snowball Earth events, sea glaciers could have failed to completely penetrate narrow marine embayments hydraulically connected to the ocean, thereby satisfying a necessary condition for creating refugia for photosynthetic life

Our previous studies considered the viability of marine embayments (referred as inland seas in previous publications) to provide refugia for photosynthetic eukaryotes during the Snowball Earth events of the Neoproterozoic [*Campbell et al.*, 2011, 2014]. We found that straight-sided marine embayments limit sea-glacier penetration, and in regions of net sublimation, an invading sea glacier could sublimate completely before reaching the end of the embayment, under a limited range of climate conditions. In Campbell et al. [2014], we considered ice flowing through a narrow entrance into a wider channel. Ice flowing through a narrow entrance moves parallel to the embayment axis. The ice then spreads out laterally into the regions immediately adjacent to the entrance [*Campbell et al.*, 2014; Figure 4]. We found that under some combinations of entrance widths and climate conditions (i.e. temperature and sublimation rate) sea-glacier-free regions could exist near the entrance of the channel. We suggested that a similar situation could exist behind an obstruction such as a promontory or island farther along the channel, creating a sea-glacier-free zone, which we call an ice shadow. Such a region might also be free of locally-grown sea ice (Figure 1). We defined sea-glacier-free zones as regions where locally grown sea ice would be thin enough to allow for transmission of light.

These ice shadows might have provided refugia for photosynthetic organisms in channels that would otherwise have been completely filled by sea glaciers. Channels containing several ice-free spots could be more robust at preserving organisms during the glaciation-deglaciation cycle of a Snowball Earth event.

Here we calculate the flow of floating ice around obstructions to address two questions: 1) how much does a promontory reduce sea-glacier penetration, and 2) can promontories generate thin-ice zones in their lee capable of independently acting as refugia?

In Section 2, we provide a theoretical framework for why ice shadows exist and provide examples of ice shadows on the modern Earth. Section 3 describes the methods we use for ice-flow modeling and how we constrain appropriate climate conditions for our model. In Section 4 we describe the results of our experiments in term of metrics useful for exploring sea-

glacier penetration. In Section 5 we discuss our results including how promontories affect sea-glacier penetration, our underlying assumptions, and model limitations. Section 6 provides our conclusions.

## 2 Modern Ice Shadows

As floating ice approaches islands, promontories or other obstructions, the ice slows down and tends to thicken relative to nearby unobstructed ice. The resulting surface gradients transverse to the flow axis allow ice to flow around and away from

the obstruction. When ice flowing down a channel encounters a constriction, the same amount of ice moving along the channel must move through a smaller cross-sectional area in the constricted region. Therefore the average ice speed is greater through the constricted region. However, near the obstruction, lateral drag slows the ice relative to ice at a corresponding position upstream or downstream from the obstruction. Due to the high effective viscosity of ice, the fast-flowing ice moving through the constricted region then has difficulty changing direction and flowing toward the downstream

side of the obstruction, creating a thin-ice zone, which we refer to as an ice shadow. In faster flow, the ice is swept farther past the promontory before it can spread significantly. A qualitative illustration of an ice shadow is shown in Figure 1, and Section 4 includes model results.

Features similar to the ice shadows that we are hypothesizing can be seen on the modern Earth. In McMurdo Sound,

Antarctica, the Ross Ice Shelf creates an ice shadow as it flows past White Island (Figure 2a). On the opposite side of White Island, the ice is thinned sufficiently to allow a tidal crack to remain open continuously. In an interesting coincidence, this ice shadow is a refugium. A colony of Weddell Seals remains isolated from other Weddell Seal populations because an ice-shelf advance isolated White Island sometime between 1947 and 1956. The tidal crack remains their only point of access into the ocean [*Gelatt et al.*, 2010].

An ice shadow in grounded ice can be seen in Taylor Valley in Antarctica (Figure 2b), where a small distributary of Taylor Glacier flows around Finger Mountain in a 180-degree turn, becoming the much-thinner Turnabout Glacier, which then incompletely penetrates Turnabout Valley.





## 3 Methods

Here we explore how introducing a square promontory into a channel affects ice flow as compared to flow in a straight-walled channel without a promontory. In our experiments, we vary three parameters: the size of the square promontory, the surface air temperature, and the sublimation rate. Surface air temperature and sublimation rate are boundary conditions on
the ice-flow model (described in Section 3.1). The ice-flow model can be used with any combination of surface air temperature and sublimation rate; however, since we are concerned with potential refugia for photosynthetic life, we choose to investigate primarily climate conditions that might suit the transmission of light through ice (see Section 3.2). These two requirements interact: warm surface temperature is needed to prevent thick ice growing locally, blocking light transmission, but warm temperature softens glacier ice, allowing it to flow more easily around the promontory where it could cover the
potential refugium.

We performed a series of experiments to determine the amount of thinning as ice flows around a promontory. The geometry of our experiments consisted of an idealized rectangular channel with width $W$ and length $L$. A square promontory (Figure 1), with side-length $L_p$, was placed along one sidewall. For most experiments, the promontory was centered at $x = 0.85L$.
We assumed that the promontory walls and the channel sidewalls were very steep and very high, and hence the sea glacier could not move onto the promontory; a low promontory that could be over-ridden by the sea glacier would require a different model. This approach is conservative because other wall configurations could increase drag and reduce sea-glacier penetration. We then solved iteratively for sea-glacier thickness $h(x,y)$, penetration length $L_g$, and the velocity field $\mathbf{u}(x,y)$, while varying promontory size $L_p$ and combinations of surface temperature $T_s$ and sublimation rate $\dot{b}$ that produced local sea
ice that was 50 m thick (see Figure 3). We used ten combinations of surface temperatures ranging from -5.7°C to -4.6°C and sublimation rates ranging from -2 to -20 mm/year. This range of climate conditions allows for thin sea ice (Section 3.2). The promontory side length $L_p$ is varied from 0 (i.e. no promontory) to 100 km.

### 3.1 Sea-glacier flow model

To simulate the behavior of a sea glacier flowing in a channel containing a promontory, we used an ice-flow model solving
an approximation to the Navier-Stokes momentum-balance equations (called the Shallow Shelf Approximation [*Morland*, 1987; *MacAyeal et al.*, 1996]) coupled to a time-independent mass conservation equation. Here we outline the ice-flow modeling procedure; *Campbell et al.* [2014] provided more details. The Shallow Shelf Approximation is an approximation of the Navier-Stokes equations that is suitable for floating ice where the vertical dimension is much smaller than the horizontal dimensions. This model incorporates both lateral shearing, and longitudinal stretching. We specify mean-annual
surface temperature and sublimation rate, both spatially uniform and temporally constant. A Glen's Flow Law viscosity is used [*Glen*, 1955] with an ice-softness parameter based on the mean-annual surface temperature [*Cuffey and Paterson*, 2010, pg. 75]. Along the entrance of the channel, a constant-pressure boundary condition is applied, representing a uniform ice





thickness along the boundary. Along the sidewalls and along the promontory, no-flow conditions are applied. Along the terminus of the sea-glacier, an integrated hydrostatic equilibrium condition is used to simulate the terminus contact with seawater. We chose not to model the effect of sea or mélange backpressure; however, doing so would reduce sea-glacier penetration. To determine penetration length $L$, we iteratively changed the sea-glacier length until the dynamic ice flux

entering the channel balanced the kinematic ice flux lost by sublimation integrated over the entire glacier surface area. These equations were solved using a commercially available finite-element solver, COMSOL Multiphysics® (comsol.com).

**3.2 Temperature and sublimation rate**

In this study, we search for combinations of surface temperature and sublimation rate that can maintain thin sea ice or open water. For our purposes, we define thin sea ice to be less than or equal to 50 m, because that is the ice-flow model's

resolution limit for thin ice. Sea ice thickness would be larger if the surface temperature was colder or if the sublimation rate was lower.

**3.2.1 Equilibrium sea-ice thickness**

Sublimation rate and surface temperature are related to equilibrium sea-ice thickness tjhrough a heat-conduction equation

$$h = \frac{k(T_f - T_s)}{\rho_i L_i \dot{b} + F_{geo}},$$ (1)

where $h$ represents the ice thickness, $k$ is the thermal conductivity, $T_f$ is the freezing temperature in the ocean, $T_s$ is the surface temperature, $\rho_i$ is the density of ice, $L_i$ is the heat of fusion of ice, $\dot{b}$ is the rate of ice growth at the base (assumed equal to the net sublimation rate in steady state; thicker, flowing ice would not require freeze-on to balance the surface sublimation), and $F_{geo}$ is the geothermal flux. The sublimation rate is linked to temperature by the Clausius-Clapeyron equation, as in *Campbell et al.* [2014]. Equation 1 is a simplification of Equation 1 of *Warren et al.* [2002] in which we have

omitted the penetration of solar radiation. Equilibrium sea-ice thicknesses calculated using Equation 1 are shown in Figure 3. By choosing not to consider the effect of solar absorption within the ice, our calculations provide an upper bound on sea-ice thickness for a given $T_s$ and $\dot{b}$. Values used to calculate $h$ in Figure 3 are given in Table 1. The albedo of ice can strongly affect sea-ice thickness with bare ice likely absorbing enough light to remain thin enough to allow photosynthesis under the ice, although photosynthesis may be possible within brine pockets of otherwise thicker ice [*Warren et al.*, 2002];

however a salt crust may develop on sea ice, thereby increasing the sea-ice albedo [*Light et al.*, 2009], increasing ice thickness, and possibly preventing photosynthesis under the ice surface.

**4 Results**

General patterns can be recognized from the ice-flow modeling experiments. For example, in Figure 4, $T_s$ = -5.33 °C, $\dot{b}$ = 12.70 mm/yr, and $L_p$ = 60 km. Far upstream of the promontory, ice flow is fastest along the center of the channel, and the




pattern of ice flow is indistinguishable from ice in flow an unobstructed channel; however the overall ice speed is slower in the obstructed case. As ice approaches the promontory, it slows, and dynamically thickens. The length scale for this longitudinal strain is ~ $L_p$. Ice thickening and longitudinal strain are not prominent toward the center of the channel; therefore ice in the center of the channel thickens only slightly upstream of the promontory. This thickness gradient between

5 ice directly upstream of the promontory and ice near the channel center directs ice flow toward the center of the channel; the location of fastest flow here is displaced toward the sidewall opposite the promontory. As the ice moves beyond the promontory, it spreads laterally, driven by a thickness gradient between the center of the channel and the region immediately downstream of the promontory where the ice is thin; here the location of fastest flow returns toward a more central position. Far downstream of the promontory, the pattern of ice flow is indistinguishable from the pattern in an unobstructed channel;

however the overall ice speed is slower in the obstructed case. Ice flow is again fastest in the center of the channel.

We use two metrics to quantify the presence of ice shadows. These are thickness drop $\Delta H$ and thin-ice percentage *TIP*. Thickness drop $\Delta H \equiv \overline{H_U} - \overline{H_D}$ where $\overline{H_U}$ and $\overline{H_D}$ represent mean ice thickness in regions directly upstream and downstream of the promontory respectively. Those upstream and downstream evaluation regions are chosen to have length $3L_p$ and width

$L_p$. (see dashed lines in Figure 4). Scaling the evaluation regions with the size of the promontory allows the thickness drop to be more completely captured than if we were to use evaluation regions of constant size independent of $L_p$, because we expect that the spatial extent of the disturbance should scale with $L_p$. Thin-ice percentage *TIP* is defined as the percentage of the downstream evaluation region where ice thickness falls below a specified value $H_{\text{thin}}$.

Thickness drop $\Delta H$ increases with increasing promontory size $L_p$ as seen in Figure 5a where $\Delta H$ is averaged over each of the 10 $T_s$ and $\dot{b}$ combinations for each $L_p$ (Section 3). A penetration percentage can be calculated, by dividing the penetration distance of a sea glacier into a channel with a promontory, by the penetration distance in a control run in which a sea glacier flows into a channel without a promontory, for the same climate variables $T_s$ and $\dot{b}$. Figure 5a shows the mean value of the penetration percentage, calculated for each of the 10 $T_s$ and $\dot{b}$ combinations for each $L_p$.

Our results reveal a similar correspondence between larger promontory sizes and higher downstream thin-ice percentages (Figure 5b). For promontories with $L_p$ of 10-60 km, centered at 0.85 $L_g$, where $L_g$ is the penetration distance in the control run, typically 28% of the ice in the downstream evaluation region is less than 75 m thick, and typically 1% is less than 50 m thick. For promontories with $L_p$ of 70-100 km, typically 94% of the ice in the downstream evaluation region is less than 75 m

thick, and typically 34% is less than 50 m thick.





## 5 Discussion

Figure 5a (blue curve) demonstrates a reduction in penetration percent as $L_p$ increases. This result is physically sensible because the promontory partially blocks the channel. Increasing $L_p$ increases the perimeter of the channel and therefore the overall drag experienced by the sea glacier. This result suggests that the choice of rectangular channels in *Campbell et al.*

[2014] represents an upper bound for sea-glacier penetration into channels, because inclusion of promontories or islands would have reduced sea-glacier penetration, consistent with the continental constriction experiments of *Tziperman et al.* [2012].

An interesting result of this study is that we can calculate a "promontory efficiency". Smaller promontories are more

efficient at increasing thickness drop and reducing overall sea-glacier penetration per sidewall unit length (Figure 5c). While a single small promontory is relatively ineffective at generating a significant thickness drop or reduction in sea-glacier penetration, it is possible that an array of small promontories could be more effective than single, larger promontory. At sufficiently close spacing, however, the promontories would interact with their neighbors, reducing this effect. This further illuminates the concept that a straight-walled channel provides an upper bound on sea-glacier penetration.

Our experiments used square promontories. By generalizing this procedure to incorporate rectangular geometries, it would be possible to parse out the relative importance of length along the channel or width across the channel at reducing sea-glacier penetration.

For our experiment, we also tested the model for sensitivity to promontory position by varying the location $X_p$ of the center of the promontory from 0.1 $L$ to 0.9 $L$, where $L$ is the total length of the glacier; Figure 6 shows that the thickness drop $\Delta H$ remains roughly constant (at ~340m) in the range of 0.1 $L$ to 0.7 $L$, and falls off sharply (from ~340 m to ~270 m) in the range of $0.7L$ to $0.9L$. Despite the more significant thickness drop in the $0.1L$ - $0.7L$ range of positions, we chose a promontory location of 0.85 $L$ in order to increase the likelihood that ice downstream in the shadow could thin sufficiently to

allow light to pass through.

It is possible that the surface of an invading sea glacier would be higher than the surface of a promontory. We assumed that the promontory walls and the channel sidewalls were steep and high, so that the sea glacier could not move onto the promontory. If the sea glacier were allowed to flow onto a shallow promontory, it might override the promontory,

eliminating an ice shadow. However, a grounded sea glacier flowing over a promontory would still tend to slow because of the additional basal resistance, and the penetration length $L$ would be decreased, increasing the probability of a refugium farther along the channel. To model this effect would require a model capable of capturing basal resistance, which would be more complex than the model used here.

In order to ensure numerical stability of our model, we enforced a minimum ice thickness of 50m. However, this is a large value compared to the thickness of ice through which light will transmit, and therefore photosynthetic life could survive. If the minimum ice thickness could be reduced, then we would be able to better quantify the areal extent of zones with light

transmission.

## 6 Conclusions

In this study we were interested in understanding the effect that promontories in channels could play in controlling the suitability of these channels as refugia for photosynthetic life during the Snowball Earth events. We calculated the flow of floating ice around promontories for conditions suitable during a Snowball Earth event. We have two major conclusions:

1) Ice shadows form behind promontories in channels, and these shadows could possibly provide refugia under the right circumstances. Ice flowing around a promontory is thinner downstream of the promontory relative to ice in a channel without a promontory; however, due to the resolution of our model, we were unable to quantitatively demonstrate if or when ice downstream of a promontory could be thin enough to provide a refugium in the ice shadow. Modern examples and analogues can be found of both grounded and floating ice that thins as it flows around obstructions. For suitable conditions

during the Snowball Earth events, sea-glacier ice could have thinned sufficiently on the downstream sides of large promontories in marine embayments to allow for photosynthesis.

2) The effect of including promontories along the shoreline of a channel reduces the invading sea glacier's total penetration, therefore potentially allowing a refugium to exist at the downstream end of the channel, over a larger range of climate conditions.

## 7 Acknowledgments

This research was supported by NSF grant ANT-11-42963.

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





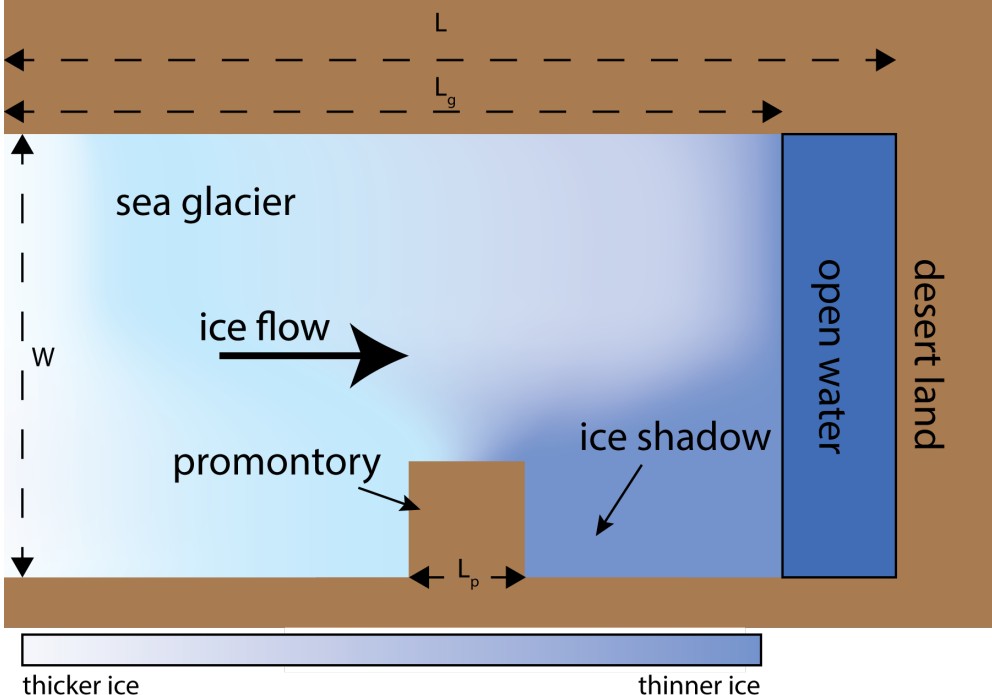

**Figure 1.** Cartoon illustration of ice shadow forming as sea glacier flows around a promontory in a channel. Note how ice thickness ahead of the promontory and thins behind the promontory.







**Figure 2. a) satellite image of Ross ice shelf flowing around White Island (78.1 °S, 167.4 °E) forming an ice shadow on the northwest side of White Island where a tidal crack can form (inset). b) Taylor Glacier flowing around Finger Mountain (77.8 °S, 161.3 °E) and incompletely penetrating Turnabout Valley. Arrows indicate the ice-flow direction. Map data: Google, DigitalGlobe**

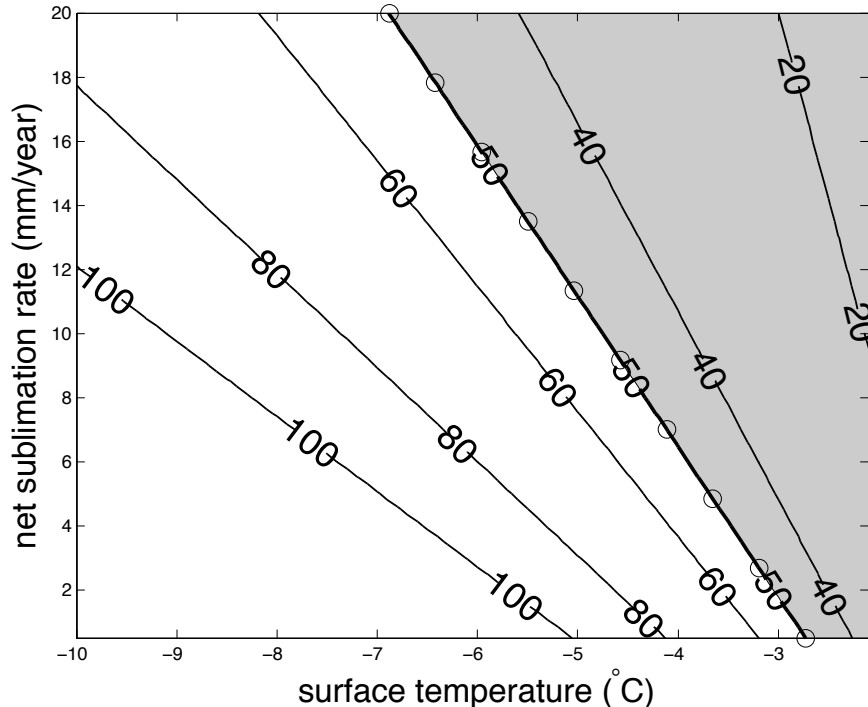

**Figure 3. Ice-thickness contours (in meters) for combinations of surface temperature and sublimation rate as predicted by Equation 1. Combinations of sublimation rate and surface temperature used in experiments are shown with hollow circles.**




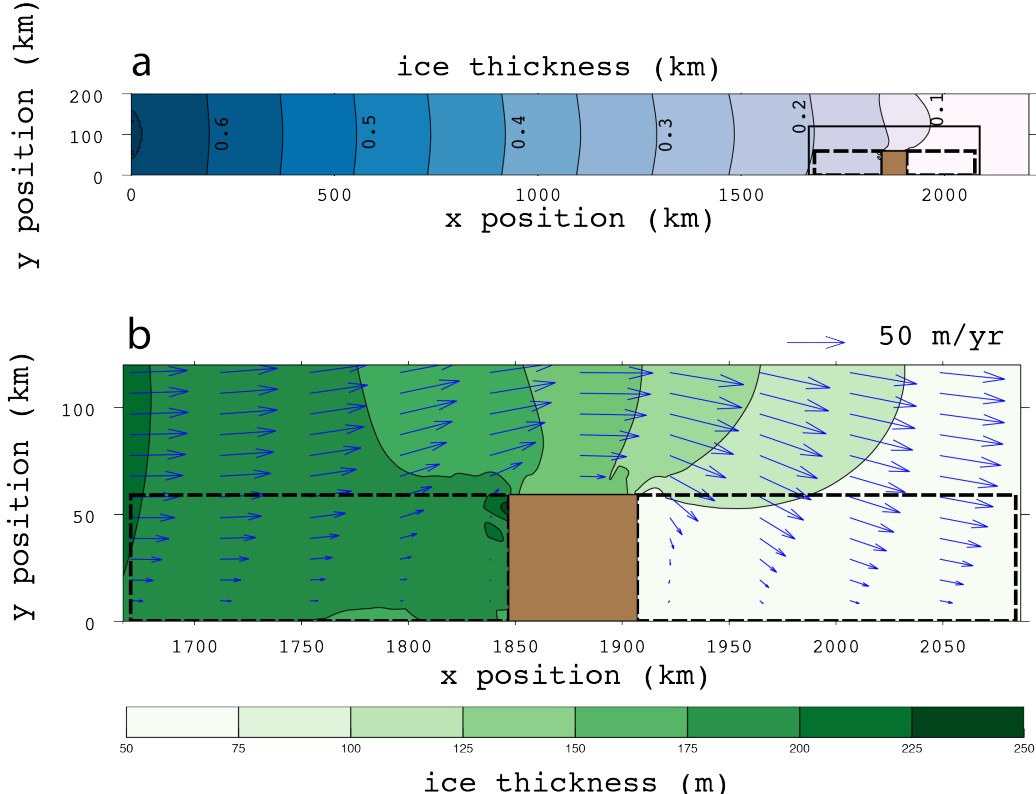

**Figure 4. Results from a sample sea-glacier flow model where $T_s$ = -5.33 °C, $\dot{b}$ = 12.70 mm/yr, $L_p$ = 60 km. Arrows indicate flow rate and direction; a reference arrow for 50 m/yr is shown. Contours show ice thickness. Dashed rectangles indicate upstream**

5 **(left) and downstream (right) regions where ice thickness is evaluated. (a) the entire model domain, with a 50-m contour interval.**

**(b) An enlarged section around the promontory, with a 25-m contour interval.**





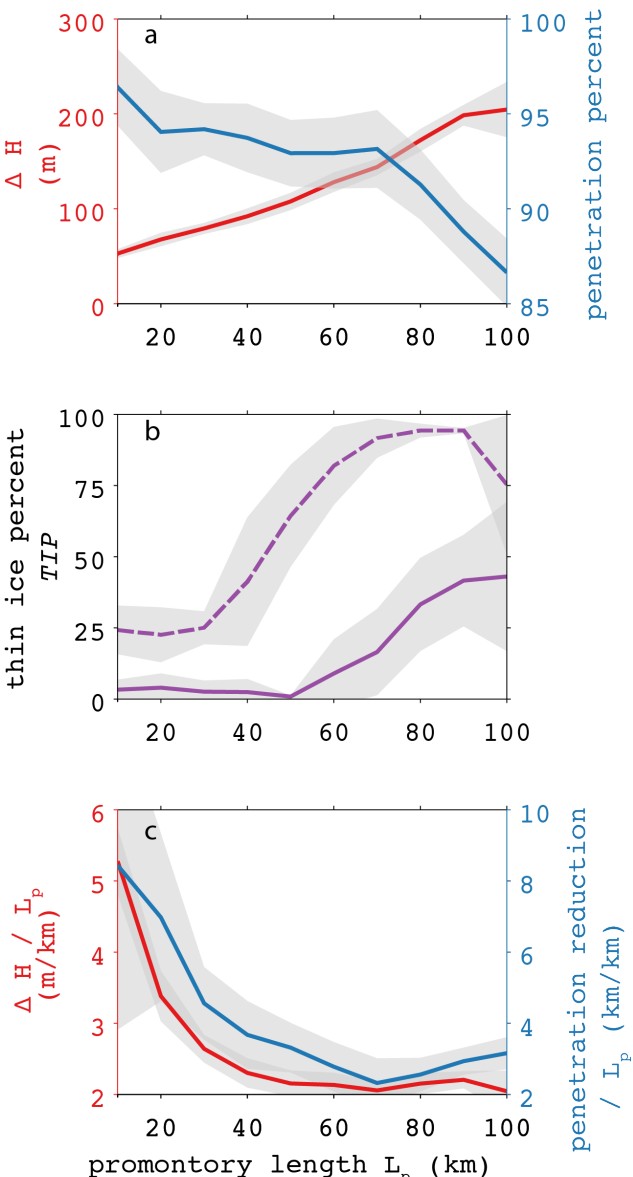



**Figure 5. (a) thickness drop $\Delta H$ (red) and penetration percent (blue) compared to a control run in a channel with no promontory.**

**(b) Thin ice percentage for $H_{\text{thin}}$=50 m (solid curve) and for $H_{\text{thin}}$=75 m (dashed curve). (c) thickness drop $\Delta H$ and penetration**

**percent scaled by promontory length $L_p$. For these experiments the channel width $W$ is 200 km. Values are averaged over all $T_s$**

**and $\dot{b}$ combinations for each promontory length $L_p$. Gray regions include one standard deviation among 10 $T_s$ and $\dot{b}$ combinations**

5 **(Section 3).**





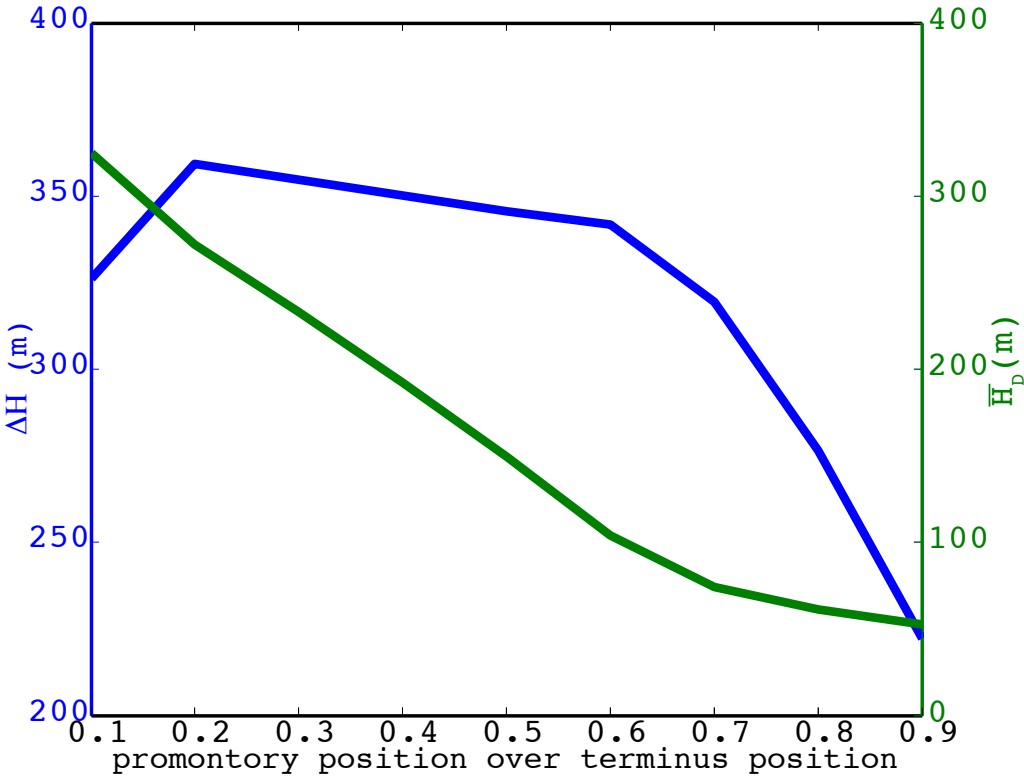

**Figure 6. Thickness drop $\Delta H$ and mean down-stream thickness $\overline{H_D}$ versus center of promontory position $X_P/L$ along sidewall, expressed as a fraction of the glacier's total penetration length. Sea-glacier terminus is to the right.**

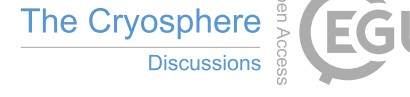

**Table 1. Constants used in analysis.**

| Name | Symbol | Value | Units |
|---|---|---|---|
| geothermal flux | $F_{geo}$ | 0.1 | W m$^{-2}$ |
| acceleration of gravity | $g$ | 9.81 | m s$^{-2}$ |
| thermal conductivity of ice | $k$ | 2.5 | W m$^{-1}$ K$^{-1}$ |
| latent heat of fusion of ice | $L_i$ | $3.3 \times 10^5$ | J kg$^{-1}$ |
| freezing temperature of sea water | $T_f$ | -2 | °C |
| ice density | $\rho_i$ | 917 | kg m$^{-3}$ |