# Peer review of "Could promontories have restricted sea-glacier penetration into marine embayments during Snowball Earth events?"

_The Cryosphere, 2016_

## Referee Comment (RC1) · Anonymous Referee #1 · 3 Nov 2016

Review for The Cryosphere: *Could promontories have restricted sea-glacier penetration into marine embayments during Snowball Earth events?*

**1   General comments**

In this study the authors explore how square obstacles modify the flow of floating ice in a channel with the objective of quantifying thin ice regions or ice shadows in which life could have persisted during snow ball Earth. The scientific question is and interest-

ing one and I believe this study has a potential to give useful insight to the proposed theories of how life could have persisted when oceans were covered by glaciers. Nevertheless, as it is, this manuscript appears to be only a small extension to Campbell et al., 2014. The methods and model used here are identical to Campbell et al., 2014 and the setup for the modeled domain is very similar as well. Specifically, the only modification to the previously studied setup is that before the upstream inflow boundary condition of constant thickness was applied where the channel is narrowed and so its interpretation was a narrow entrance to an embayment. In the current setup the upstream inflow boundary condition is moved further up past the narrowing and so the interpretation of the narrowing inside of the channel is that it is a promontory. This shift of a narrowing further inland and shift of the inflow boundary condition further upstream is what supposedly allows the current ice shadows to possibly persist unlike before. But it seems that it really just comes down to how far away from the narrowing the ocean body lies, if that is the case, it seems that such conclusion could have been reached without further modeling (especially since the ocean is not included in the model). I think for this study to provide some new insight that has not been shown in Campbell et al., 2014 yet, further extension should be included possibly focusing on one of the following questions mentioned in the manuscript, but not elaborated on:

1) It is unclear why authors chose to use a model which has such high minimum thickness requirement, given their main motivation is to answer the question weather life could have persisted in ice shadows of much smaller thickness than allowed by this particular model. There are many other models available that solve the shallow shelf approximation and that allow for much smaller thickness. Using a more suitable model thus could make it possible to answer the question of the aerial extent of zones with light transmission, which the authors express the pity not to be able to answer. Further, it may be worth validating the model with the examples of current climate that are provided in the manuscript. Applying the validated model to known specific embayments from the past and evaluating where life could have persisted would be of great use, thought the later is probably past the scope of this study.

[Figure]

2) In case the main motivation is not to exhaustively answer the questions regarding ice shadows as refugee for life, but to study the flow of floating ice past obstacles, this study should be a bit more comprehensive and less hand wavy. For example, it would be useful and interesting to analyze for which relative size of obstacle to channel width can a model which uses floating ice only be used as an upper bound for sea glacier penetration length (see specific comments). The authors claim but do not show that this model can always be used as an upper bound. Also the discussion of promontory efficiency should be more elaborate and the speculation that series of small promontories could be more efficient than a large one should be modeled and analyzed rather than speculated about.

**2  Specific comments**

P2_L21-23: A finding from previous work of the authors (Campbell et al., 2014) is used as a motivation for further work. These ice free zones formed on the sides at narrow entrances to embayments were however concluded not to be suitable as refugee: *Sea-glacier-free zones near the channel entrance would grow thick ice locally if they are located in a cold region of the inland sea. We conclude that none of the sea-glacier-free zones observed in our models near the channel entrance would act as refugia, because they are only observed with colder temperatures that would locally generate thick sea-ice.* Is moving the obstacles away from the open ocean rather than keeping them at the entrance where they would not persist because of the closeness of the ocean body the only thing that distinguishes this work from the previous article? If not, some discussion would be useful regarding of why the obstacles considered in this paper could persist while those in the previous one could not. How far in the embayment does the promontory need to be located and is $x = 0.85L$ far in enough to be shielded from the main body of the ocean? For what ratio of $L$ to $W$ is $x = 0.85L$ a suitable choice? I would suggest elaborating on and justifying some of these choices.

P3_L2: Section 2 does not provide theoretical framework, it only provides intuitive explanation of the mechanisms involved.

P3_L20-21 & Fig 2a: It would be useful to show thickness contours superimposed on top of or even instead of the satellite image. From the simple photograph it is not obvious where exactly is the ice shadow and how significant it is.

P3_L26-28 & Fig 2b: This example doesn't seem relevant to this study which only considers floating ice, while this is a grounded ice example.

P4_L17 & P4_L17: For consistency, give values of $W$ and $L$ used in the model and/or express $L_p$ considered in terms of $W$ or $L$.

P4_L17 & P7_L30-32: Can you show that the following statements are always true? If not, then you cannot speak of an upper bound or conservative approach to be guaranteed by this model unless specifying when exactly it is true:

P4_L17: *This approach is conservative because other wall configurations could increase drag and reduce sea-glacier penetration.*

P7_L30-32: *However, a grounded sea glacier flowing over a promontory would still tend to slow because of the additional basal resistance, and the penetration length L would be decreased, increasing the probability of a refugium farther along the channel.*

I suspect that as $L_p \to W$ there could be a point when $L_g$ reached by flow through the narrow opening will be smaller than $L_g$ reached by flow over the promontory.

P4_L24-26: This sentence is a bit strange, just say that you solve the Shallow Shelf Approximation in steady state, later you repeat what it is anyways.

P5_L1: By no-flow conditions do the authors mean no-slip? How do you justify this choice in the context of wanting to be conservative in computing $L_g$ as was emphasized before?

P5_L4: Penetration length was previously called $L_g$ not $L$

P5_L9: Why is the minimum thickness requirement so high? It looks like in your previous study 5 m thick ice was possible. Why not use then a realistic thickness that allows for sufficient light penetration?

P5_L13: typo - through in stead of tjhrough

P5_L13: keep consistent terminology, keep clear when referring to sea ice and when to sea glacier.

P5_L22-26: The discussion on albedo seems a bit out of place given solar radiation is not included.

P5_L28-29: Reformulating the first two sentences may be helpful. First statement is completely generic. Second sentence is just a fragment and it is unclear what it connects to.

P6_L2-3: Does this length scale come out of the equations or is it just by eye comparison?

P6_L12: The naming of this metric as thickness drop is a bit deceiving as it includes both thickness increase and thickness decrease so it is counting the same thing twice. Perhaps a better way to quantify thickness drop would be to compare the thickness past the obstacle to the thickness in the control run. With the metric as it is a hypothetic situation of ice thickness increasing upstream from the obstacle and no change of the flow showing downstream would show as a thickness drop. Using a more general metric would be useful for example to compare thickness drop for the case of narrow entrances to an embayment as in Campbell et al., 2014.

P6_L26-30 & Fig 5b: This paragraph suggest there are two regimes, 10-60 km and 70-100 km, however the figure shows rather smooth transition.

P7Ł9: What happens to the efficiency of promontories as defined here when $L_p \to 0$ and does it make sense?

P7_L11-14: I would suggest this effect of an array of promontories and the effect of spacing between them to be included in this paper for completeness, rather then hypothesizing about it.

P7_L16-18: It seems that this study would be a right place to include this generalization to rectangular geometries.

P8_L2-5: Perhaps mention earlier the reason you keep using 50 m for minimum thickness - this choice was not justified earlier and seemed strange. Also give reasonable values of ice thickness for light transmission to be sufficient for life to persist.

P8_L14: *Modern examples and analogues can be found of both grounded and floating ice that thins as it flows around obstructions* - is not really a 'conclusion' of this study.

Fig 4: flow speed in stead of flow rate

Fig 5: Why is there a sudden drop for the 75m thin ice percentage for $L_p = 100$km? Does the trend continue for higher $L_p$ and if so what is the reason for the reversal of the trend?

---

## Referee Comment (RC2) · S. L. Cornford (Referee) · 29 Nov 2016

I have to confess to not knowing much about Snowball Earth events. That said, the introduction makes it clear what the paper is about: if colonies of photosynthetic organisms survived these events, then they must have been exposed to sufficient light, so that sea glacier coverage must have not been complete. That idea has been considered before (by the same authors, and by Pollard 2005): the innovation here is the a look at the inclusion of partial obstacles in channels that would otherwise be covered with thick ice.

The term 'sea glacier' was one I did not know, but a quick look at Pollard 2005 (also about the survival of photosynthetic organisms) credits the term to one of the authors

here (Warren) and refers to ice that, although floating and formed by freezing of the sea, looks more like modern land ice in terms of thickness and salinity. Like Pollard 2005, this paper then assumes dynamics similar to modern ice shelves and tongues. It proposes that rocky promontories, by modifying the flow field create an ice shadow, ie a region of thin ice downstream from the promontory, thin enough for photosynthesis to take place beneath the ice.

This is a modelling study, which is looking at the thickness of ice in the shadow with a numerical model based on the Shallow Shelf ice flow approximation (conventionally used to study ice shelves and tongues), and concludes that the mechanism is plausible. It does make reference to some modern ice shadows, just to be clear that we a are talking about a genuine physics phenomenon (if not necessarily a genuine biology phenomenon)

Coming from a numerical modelling perspective largely centred on contemporary ice dynamics, I enjoyed this paper, but thought it was too short. I would have like to have seen some fleshing out of the 'more than one promontory' discussion with model results, though I don't imagine the conclusion would be very different. Likewise, I think some runs exploring the boundary conditions would be instructive - what if the promontory does not impose zero tangential flow (Dirichlett BC), but finite draq (Robin BC)? Does the resulting variation in ice shadow suggest that refugia would be common or rare?

That aside, it is interesting to see the present interest in flow fields with lateral variation having an impact in thinking about the distant past. I suppose some might describe the paper as a bit speculative, but the dynamical model is well founded and the discussion is clear enough for the most part.

Specific Comments ———————————

P1, L21 : Neoproterozoic – how about adding a time period?

P3, L9: It is not just surface gradients around the obstacle that changes the flow. Even a flat ice mass would see its flow deformed, through the interaction between viscous stress in the ice and the no-normal flow (and no/reduced tangential flow) at the obstacle wall.

P3,L10. This sentence assumes x,y-incompressible flow, which is not quite correct. The flow is incompressible but you must take the z-component into account, so e.g the same volume of ice can be moved through a constriction at the same speed as up and down stream if is is thicker within the constriction.

P4,L15: 'This approach is conservative...' : I don't see what you mean here.Are you just saying that the wall geometry is a sensible idealized case, or something else?

P4:L29 '...iteratively... produced local sea ice that was 50m thick (fig 3). This is confusing. I think you are picking points on the h = 50 m contour from eq 1 (so choosing pairs of surface temperature Ts and sublimation rate b), then computing flow model solutions Lg, u(x,y), h(x,y) for a variety. The iteration is just how the ice flow model solves its PDEs? But I had to read to sec 3.2.1 to realize this.

P4: L25. Normally just an approximation to the Stokes equations (but OK, the Stokes are an approximation to the Navier-Stokes)

P5, L3: 'Integrated hydrostatic equilibrium...' This is the normal shelf front boundary condition, yes? In which case it includes the sea pressure.

P5,L9: 'thin ice < 50 m' . This isn't really a resolution limit, because the model doesn't have a vertical resolution. Presumably, it is related to solver stability (e.g a region of thick ice surrounded by thin ice starts to look like an elliptic PDE with Neumann conditions an all boundaries)

P6,L2: slower, given the same channel width outside the promontory? i.e having the promontory just makes the channel look narrower far upstream.

P6,L4: 'thickness gradient... directs'. Not entirely - the stress balance and BC's alone

would produce this deflection for uniform h(x,y) (see earlier comment). The thickness is a result of the flow as much as the other way round.

P6: Fig 5c is not discussed. What does it add?

P7, L10 (promontory efficiency paragraph). Seems a bit too vague. Why not do some runs that explore this idea, if you are determined to discuss it. I'm sure it is true that a modulated wall exerts more net drag coefficient than a straight one.

---

## Author Comment (AC1) · 15 Mar 2017

In this study the authors explore how square obstacles modify the flow of floating ice in a channel with the objective of quantifying thin ice regions or ice shadows in which life could have persisted during snow ball Earth. The scientific question is and interesting one and I believe this study has a potential to give useful insight to the proposed theories of how life could have persisted when oceans were covered by glaciers. Nevertheless, as it is, this manuscript appears to be only a small extension to Campbell et al., 2014. The methods and model used here are identical to Campbell et al., 2014 and the setup for the modeled domain is very similar as well. Specifically, the only modification to the previously studied setup is that before the upstream inflow boundary condition of constant thickness was applied where the channel is narrowed and so its interpretation was a narrow entrance to an embayment. In the current setup the upstream inflow boundary condition is moved further up past the narrowing and so the interpretation of the narrowing inside of the channel is that it is a promontory. This shift of a narrowing further inland and shift of the inflow boundary condition further upstream is what supposedly allows the current ice shadows to possibly persist unlike before. But it seems that it really just comes down to how far away from the narrowing the ocean body lies, if that is the case, it seems that such conclusion could have been reached without further modeling (especially since the ocean is not included in the model). I think for this study to provide some new insight that has not been shown in Campbell et al., 2014 yet, further extension should be included possibly focusing on one of the following questions mentioned in the manuscript, but not elaborated on:

1) It is unclear why authors chose to use a model which has such high minimum thickness requirement, given their main motivation is to answer the question weather life could have persisted in ice shadows of much smaller thickness than allowed by this particular model. There are many other models available that solve the shallow shelf approximation and that allow for much smaller thickness. Using a more suitable model thus could make it possible to answer the question of the aerial extent of zones with light transmission, which the authors express the pity not to be able to answer. Further, it may be worth validating the model with the examples of current climate that are pro- vided in the manuscript. Applying the validated model to known specific embayments from the past and evaluating where life could have persisted would be of great use, thought the later is probably past the scope of this study. **The model we use here has solver stability issues at thicknesses less than 50m. We have spent significant time trying lower this limitation. Because of the nature of the Shallow Shelf Equations (SSA), instability is introduced whenever thickness is allowed to become negative (or zero). We deal with this by setting thickness back up to a reasonable value. With this problem ice is being sublimated and stretches downstream, both of these conditions ensure that ice will thin to zero with a suf-**

[Figure]

**ficiently long channel. Any solver attempting to solve the SSA will have stability issues at some thickness with this problem setup.**

2) In case the main motivation is not to exhaustively answer the questions regarding ice shadows as refugee for life, but to study the flow of floating ice past obstacles, this study should be a bit more comprehensive and less hand wavy. For example, it would be useful and interesting to analyze for which relative size of obstacle to channel width can a model which uses floating ice only be used as an upper bound for sea glacier penetration length (see specific comments). The authors claim but do not show that this model can always be used as an upper bound. Also the discussion of promontory efficiency should be more elaborate and the speculation that series of small promontories could be more efficient than a large one should be modeled and analyzed rather than speculated about. **As a first study in this topic, we agree that we have not exhaustively examined the effect of size, geometry, position, and number of obstacles. We agree with you about steep-walled promontories being an upper bound and have deleted this claim (see comment below). We are clear that the claim of the series of small promontories being more efficient than a large one is a speculation.**

**2 Specific comments** P2 L21-23: A finding from previous work of the authors (Campbell et al., 2014) is used as a motivation for further work. These ice free zones formed on the sides at narrow entrances to embayments were however concluded not to be suitable as refugee: Sea- glacier-free zones near the channel entrance would grow thick ice locally if they are located in a cold region of the inland sea. We conclude that none of the sea-glacier- free zones observed in our models near the channel entrance would act as refugia, because they are only observed with colder temperatures that would locally generate thick sea-ice. Is moving the obstacles away from the open ocean rather than keeping them at the entrance where they would not persist because of the closeness of the ocean body the only thing that distinguishes this work from the previous article? If not, some discussion would be useful regarding of why the obstacles considered in this paper could persist while those in the previous one could not. How far in the embayment does the promontory need to be located and is x = 0.85L far in enough to be shielded from the main body of the ocean? For what ratio of L to W is x = 0.85L a suitable choice? I would suggest elaborating on and justifying some of these choices. **There are two reasons we use a promontory in the channel that differ from the previous study with a restricted entrance, 1) we wanted to capture both the upstream and downstream effects of the promontory, the previous study only captured downstream effects, 2) ice thickness near the ocean side of the channel was too large to thin ice sufficiently to allow transmission of light except in very cold cases. We wanted to know if there was a promontory in the channel, could ice thin sufficiently at warmer temperatures.** "We found ice shadows could only exist near the entrance of the channel at very cold temperatures. In this study, we examine promontories along the channel sidewalls that are far from the entrance of the channel in order to capture both upstream and downstream effects of promontories and to determine if ice shadows can form downstream at warmer temperatures."

P3 L2: Section 2 does not provide theoretical framework, it only provides intuitive explanation of the mechanisms involved. **Changed for clarity** "In Section 2, we provide an explanation for why ice shadows exist and provide examples of ice shadows on the modern Earth. "

P3 L20-21  Fig 2a: It would be useful to show thickness contours superimposed on top of or even instead of the satellite image. From the simple photograph it is not obvious where exactly is the ice shadow and how significant it is. **I modified the figure to show figure direction and magnitude, which I think demonstrates with point.**

P3 L26-28  Fig 2b: This example doesn't seem relevant to this study which only considers floating ice, while this is a grounded ice example. **Indeed this study does only consider floating ice, and this is a grounded example. This example in Fig 2b still has merit because it shows clearly ice thinning as it moves around an obstruction. I have changed the wording to make that fact that Turnabout Glacier is**

**grounded more clear.** "The grounded Taylor Glacier flowing around Finger Mountain (77.8 °S, 161.3 °E) and incompletely penetrating Turnabout Valley. Arrows indicate the ice-flow direction."

P4 L17  P4 L17: For consistency, give values of W and L used in the model and/or express Lp considered in terms of W or L. **The term L does not explicitly enter into our model and is only used to illustrate that open water conditions exist past the end of the sea glacier. Lg is solved for in our model. We have specified W.** "The geometry of our experiments consisted of an idealized rectangular channel with width W and length L, here W is 200 km and L is long enough to prevent the sea glacier from contacting the end of the channel."

P4 L17  P7 L30-32: Can you show that the following statements are always true? If not, then you cannot speak of an upper bound or conservative approach to be guaranteed by this model unless specifying when exactly it is true:

P4 L17: This approach is conservative because other wall configurations could increase drag and reduce sea-glacier penetration.

P7 L30-32: However, a grounded sea glacier flowing over a promontory would still tend to slow because of the additional basal resistance, and the penetration length L would be decreased, increasing the probability of a refugium farther along the channel. I suspect that as Lp → W there could be a point when Lg reached by flow through the narrow opening will be smaller than Lg reached by flow over the promontory. **You are absolutely correct, our usage of upper bound was not accurate. We have changed this text on page 4.** "This approach is an idealization, if the sea glacier were allowed to move onto the promontory, it could increase or decrease sea-glacier penetration."

P4 L24-26: This sentence is a bit strange, just say that you solve the Shallow Shelf Approximation in steady state, later you repeat what it is anyways. **Changed for clarity** "To simulate the behavior of a sea glacier flowing in a channel containing a promontory,

we used an ice-flow model solving an approximation to the Stokes momentum-balance equations (called the Shallow Shelf Approximation [Morland, 1987; MacAyeal et al., 1996]) in steady state."

P5 L1: By no-flow conditions do the authors mean no-slip? How do you justify this choice in the context of wanting to be conservative in computing Lg as was emphasized before? **Changed no-flow to no-slip. This is justified because the ice is well below the melting point.** "Along the sidewalls and along the promontory, no-slip conditions are applied, a suitable condition for ice below the melting point."

P5 L4: Penetration length was previously called Lg not L **Corrected Typo. Thank you.**

P5 L9: Why is the minimum thickness requirement so high? It looks like in your previous study 5 m thick ice was possible. Why not use then a realistic thickness that allows for sufficient light penetration? **We have issues with solver stability with ice thickness less than 50m. We changed the wording to me this more clear.** "For our purposes, we define thin sea ice to be less than or equal to 50 m, because that is the ice-flow model's solver stability limit for thin ice."

P5 L13: typo - through in stead of tjhrough **Corrected Typo. Thank you.**

P5 L13: keep consistent terminology, keep clear when referring to sea ice and when to sea glacier. **Done**

P5 L22-26: The discussion on albedo seems a bit out of place given solar radiation is not included. **We wanted to be clear how this is different from Warren 2002**

P5 L28-29: Reformulating the first two sentences may be helpful. First statement is completely generic. Second sentence is just a fragment and it is unclear what it connects to. **I deleted those awkward sentences and changed the following one.** "Far upstream of the promontory, ice flow is fastest along the center of the channel, and the pattern of ice flow is indistinguishable from ice in flow an unobstructed channel (Figure 4 for an representative case);"

P6 L2-3: Does this length scale come out of the equations or is it just by eye compari-son? **This comes from a scaling argument from Kamb and Echelmeyer 1986. I've added the reference.**

P6 L12: The naming of this metric as thickness drop is a bit deceiving as it includes both thickness increase and thickness decrease so it is counting the same thing twice. Perhaps a better way to quantify thickness drop would be to compare the thickness past the obstacle to the thickness in the control run. With the metric as it is a hypothetic situation of ice thickness increasing upstream from the obstacle and no change of the flow showing downstream would show as a thickness drop. Using a more general metric would be useful for example to compare thickness drop for the case of narrow entrances to an embayment as in Campbell et al., 2014. **I do agree that this metric includes both upstream thickening and downstream thinning. However it is not intuitive to compare to case with no promontory. The no promontory case has a different overall glacier length; so comparing the same distance along the x-axis would not be a genuine comparison either.**

P6 L26-30 Fig 5b: This paragraph suggest there are two regimes, 10-60 km and 70-100 km, however the figure shows rather smooth transition. **There is a regime where TIP in Figure 5b is close to 0 and a rather smooth increase after that. I have re-worded this for clarity.** "For promontories with Lp of 10-30 km, centered at 0.85 Lg, where Lg is the penetration distance in the control run, typically 25

P7Ł9: What happens to the efficiency of promontories as defined here when Lp → 0 and does it make sense? **Certainly a zero length promontory would not reduce sea-glacier penetration in the way I have discussed. There likely exists some limit to this efficiency, as suggested by the change in inflection of the blue line on Figure 5c.** "Furthermore there is likely some limit to this effect at sufficiently small promontory sizes. "

P7 L11-14: I would suggest this effect of an array of promontories and the effect of

spacing between them to be included in this paper for completeness, rather then hypothesizing about it. **We agree that would be a good addition to the paper. However the work presented here has sufficient merit as a first study.**

P7 L16-18: It seems that this study would be a right place to include this generalization to rectangular geometries. **We agree that would be a good addition to the paper. However the work presented here has sufficient merit as a first study.**

P8 L2-5: Perhaps mention earlier the reason you keep using 50 m for minimum thickness - this choice was not justified earlier and seemed strange. Also give reasonable values of ice thickness for light transmission to be sufficient for life to persist. **We have issues with solver stability with ice thickness less than 50m. We changed the wording to me this more clear.** "For our purposes, we define thin sea ice to be less than or equal to 50 m, because that is the ice-flow model's solver stability limit for thin ice."

P8 L14: Modern examples and analogues can be found of both grounded and floating ice that thins as it flows around obstructions - is not really a 'conclusion' of this study. **True. Removed**

Fig 4: flow speed in stead of flow rate **done**

Fig 5: Why is there a sudden drop for the 75m thin ice percentage for Lp = 100km? Does the trend continue for higher Lp and if so what is the reason for the reversal of the trend? **We only explored Lp up 100 km. I imagine there will some limits to how well the model work at larger Lp. We find Lg so that the sublimation over the channel balances ice inflow into the channel. However this there is an asymmetry to ice flow generated by the promontory. This makes it so that the edge of the sea glacier will not be perfectly oriented with this model domain. I believe this problem gets worse as Lp becomes larger.**

[Figure]

Fig. 1. Figure 2. a) Satellite image of Ross ice shelf flowing around White Island (78.1 °S, 167.4°E) forming an ice shadow on the northwest side of White Island where a tidal crack can form (inset); colorba

---

## Author Comment (AC2) · 15 Mar 2017

I have to confess to not knowing much about Snowball Earth events. That said, the introduction makes it clear what the paper is about: if colonies of photosynthetic organisms survived these events, then they must have been exposed to sufficient light, so that sea glacier coverage must have not been complete. That idea has been considered before (by the same authors, and by Pollard 2005): the innovation here is the a look at the inclusion of partial obstacles in channels that would otherwise be covered with thick ice.

The term 'sea glacier' was one I did not know, but a quick look at Pollard 2005 (also about the survival of photosynthetic organisms) credits the term to one of the authors

here (Warren) and refers to ice that, although floating and formed by freezing of the sea, looks more like modern land ice in terms of thickness and salinity. Like Pollard 2005, this paper then assumes dynamics similar to modern ice shelves and tongues. It proposes that rocky promontories, by modifying the flow field create an ice shadow, ie a region of thin ice downstream from the promontory, thin enough for photosynthesis to take place beneath the ice.

This is a modelling study, which is looking at the thickness of ice in the shadow with a numerical model based on the Shallow Shelf ice flow approximation (conventionally used to study ice shelves and tongues), and concludes that the mechanism is plausible. It does make reference to some modern ice shadows, just to be clear that we a are talking about a genuine physics phenomenon (if not necessarily a genuine biology phenomenon)

Coming from a numerical modelling perspective largely centred on contemporary ice dynamics, I enjoyed this paper, but thought it was too short. I would have like to have seen some fleshing out of the 'more than one promontory' discussion with model results, though I don't imagine the conclusion would be very different. Likewise, I think some runs exploring the boundary conditions would be instructive - what if the promontory does not impose zero tangential flow (Dirichlett BC), but finite draq (Robin BC)? Does the resulting variation in ice shadow suggest that refugia would be common or rare?

That aside, it is interesting to see the present interest in flow fields with lateral variation having an impact in thinking about the distant past. I suppose some might describe the paper as a bit speculative, but the dynamical model is well founded and the discussion is clear enough for the most part.

Specific Comments ————————– P1, L21 : Neoproterozoic – how about adding a time period? **Done.** "Neoproterozoic ( 1,000 – 550 Ma)"

P3, L9: It is not just surface gradients around the obstacle that changes the flow. Even

a flat ice mass would see its flow deformed, through the interaction between viscous stress in the ice and the no-normal flow (and no/reduced tangential flow) at the obstacle wall. **Changed wording** "The resulting surface gradients transverse to the flow axis and interaction with sidewalls allow ice to flow around and away from the obstruction."

P3,L10. This sentence assumes x,y-incompressible flow, which is not quite correct. The flow is incompressible but you must take the z-component into account, so e.g the same volume of ice can be moved through a constriction at the same speed as up and down stream if is is thicker within the constriction. **Good point.** "When ice flowing down a channel encounters a constriction, ice must thicken and change its surface gradient to allow the same amount of ice to move through the smaller cross-sectional area in the constricted region. "

P4,L15: 'This approach is conservative...' : I don't see what you mean here.Are you just saying that the wall geometry is a sensible idealized case, or something else? **You are absolutely correct, our usage of upper bound was not accurate. We have changed this text on page 4.** "This approach is an idealization, if the sea glacier were allowed to move onto the promontory, it could increase or decrease sea-glacier penetration."

P4:L29 '...iteratively. . . produced local sea ice that was 50m thick (fig 3). This is confusing. I think you are picking points on the h = 50 m contour from eq 1 (so choosing pairs of surface temperature Ts and sublimation rate b), then computing flow model solutions Lg, u(x,y), h(x,y) for a variety. The iteration is just how the ice flow model solves its PDEs? But I had to read to sec 3.2.1 to realize this. **I hopefully have cleared up this confusing sentence.** "We then numerical solved for sea-glacier thickness h(x,y), and the velocity field u(x,y), searching for an Lg such that the dynamic flow of ice into the channel was balanced by sublimation over the sea glacier. We performed this operation while varying promontory size Lp and combinations of surface temperature Ts and sublimation rate b ÌĞ. The promontory side length Lp is varied from 0 (i.e. no promontory) to 100 km. We used ten combinations of surface

temperatures ranging from -5.7°C to 4.6°C and sublimation rates ranging from -2 to -20 mm/year, which produced thin (see Section 3.2), locally-growth sea ice with a thickness of 50 m (see Figure 3)."

P4: L25. Normally just an approximation to the Stokes equations (but OK, the Stokes are an approximation to the Navier-Stokes) **It is now Stokes, instead of Navier-Stokes** "approximation to the Stokes momentum-balance equations"

P5, L3: 'Integrated hydrostatic equilibrium. . . ' This is the normal shelf front boundary condition, yes? In which case it includes the sea pressure. Made this more clear hopefully. "Along the terminus of the sea-glacier, an integrated hydrostatic equilibrium condition specifies pressure due to seawater."

P5,L9: 'thin ice < 50 m' . This isn't really a resolution limit, because the model doesn't have a vertical resolution. Presumably, it is related to solver stability (e.g a region of thick ice surrounded by thin ice starts to look like an elliptic PDE with Neumann conditions an all boundaries) **You're correct. I've made this more clear** "For our purposes, we define thin sea ice to be less than or equal to 50 m, because that is the ice-flow model's solver stability limit for thin ice."

P6,L2: slower, given the same channel width outside the promontory? i.e having the promontory just makes the channel look narrower far upstream. **Changed for clarity** "Far upstream of the promontory, ice flow is fastest along the center of the channel, and the pattern of ice flow is indistinguishable from ice in flow an unobstructed channel; however the overall ice speed is slower in the obstructed case, making the channel look narrower far upstream."

P6,L4: 'thickness gradient. . . directs'. Not entirely - the stress balance and BC's alone would produce this deflection for uniform h(x,y) (see earlier comment). The thickness is a result of the flow as much as the other way round. **Changed for clarity** "This thickness gradient between ice directly upstream of the promontory and ice near the channel center, results in ice flow being directed, through a stress balance, toward the

center of the channel; the location of fastest flow here is displaced toward the sidewall opposite the promontory."

P6: Fig 5c is not discussed. What does it add? **This is discussed in the promontory efficiency paragraph.**

P7, L10 (promontory efficiency paragraph). Seems a bit too vague. Why not do some runs that explore this idea, if you are determined to discuss it. I'm sure it is true that a modulated wall exerts more net drag coefficient than a straight one. **As a first study in this topic, we agree that we have not exhaustively examined the effect of size, geometry, position, and number of obstacles. We are clear that the claim of the series of small promontories being more efficient than a large one is a speculation.**